

# Molecular systematics of the subfamily Limenitidinae (Lepidoptera: Nymphalidae)

Bidur Dhungel[1] and Niklas Wahlberg[2]

[1] Southwestern Centre for Research and PhD Studies, Kathmandu, Nepal
[2] Department of Biology, Lund University, Lund, Sweden

## ABSTRACT

We studied the systematics of the subfamily Limenitidinae (Lepidoptera: Nymphalidae) using molecular methods to reconstruct a robust phylogenetic hypothesis. The molecular data matrix comprised 205 Limenitidinae species, four outgroups, and 11,327 aligned nucleotide sites using up to 18 genes per species of which seven genes (CycY, Exp1, Nex9, PolII, ProSup, PSb and UDPG6DH) have not previously been used in phylogenetic studies. We recovered the monophyly of the subfamily Limenitidinae and seven higher clades corresponding to four traditional tribes Parthenini, Adoliadini, Neptini, Limenitidini as well as three additional independent lineages. One contains the genera *Harma* + *Cymothoe* and likely a third, *Bhagadatta*, and the other two independent lineages lead to *Pseudoneptis* and to *Pseudacraea*. These independent lineages are circumscribed as new tribes. Parthenini was recovered as sister to rest of Limenitidinae, but the relationships of the remaining six lineages were ambiguous. A number of genera were found to be non-monophyletic, with *Pantoporia*, *Euthalia*, *Athyma*, and *Parasarpa* being polyphyletic, whereas *Limenitis*, *Neptis*, *Bebearia*, *Euryphura,* and *Adelpha* were paraphyletic.

## INTRODUCTION

The butterfly family Nymphalidae has been the subject of intensive research in many fields of biology over the decades. However, the higher classification of the family is still being worked on, with the delineation of subfamilies being established fairly recently (*Wahlberg et al., 2009*). It is now clear that there are 12 subfamilies that are well supported by both molecular (*Brower, 2000*; *Wahlberg, Weingartner & Nylin, 2003*; *Wahlberg et al., 2009*) and morphological data (*Freitas & Brown, 2004*). These subfamilies have been accepted by most of the community working on Nymphalidae. The relationships of major lineages within subfamilies are now under scrutiny, with work at the level of subfamily already done on Apaturinae (*Ohshima et al., 2010*), Libytheinae (*Kawahara, 2009*), Nymphalinae (*Wahlberg, Brower & Nylin, 2005*) and Satyrinae (*Peña et al., 2006*), as well as a multitude of studies looking at relationships at lower levels within subfamilies. Here we turn our attention to Limenitidinae, a subfamily with a complex taxonomic history.

Corresponding author
Niklas Wahlberg,
niklas.wahlberg@biol.lu.se

The rank and position of Limenitidinae has always been unstable and long debated among researchers. Popularly known as a "trash can" subfamily, Limenitidinae has included groups of species that could not be placed in any recognized subfamilies and were thus retained in the subfamily just for convenience (*Harvey, 1991*; *Neild, 1996*; *Brower, 2000*). Historically, Limenitidinae were placed as a tribe in the subfamily Nymphalinae (*Smart, 1975*). Later, *Harvey (1991)* placed Limenitidinae as the tribe Limenitidini in the subfamily Limenitidinae (*sensu* Harvey) but together with three unrelated tribes Coloburini (*sensu* Harvey), Biblidini (*sensu* Harvey), Cyrestidini (*sensu* Harvey), and two genera *Pseudergolis* and *Stibochiona* (now in the subfamily Pseudergolinae). Limenitidinae (*sensu* Harvey) is equivalent to *Müller's (1886)* group III together with Cyrestidini (*Harvey, 1991*). Molecular work has finally unambiguously delineated the subfamily Limenitidinae (*Brower, 2000*; *Wahlberg, Weingartner & Nylin, 2003*; *Wahlberg et al., 2009*). Based on molecular data, the subfamily Limenitidinae is equivalent to the tribe Limenitidini of Harvey, it is sister to the subfamily Heliconiinae and does not include the taxa Cyrestidinae, Biblidinae, and Pseudergolinae (*Brower, 2000*; *Wahlberg, Weingartner & Nylin, 2003*; *Freitas & Brown, 2004*; *Wahlberg et al., 2009*).

As it is currently delineated, the subfamily Limenitidinae (Lepidoptera: Nymphalidae) comprises a little over 800 species placed in 46 genera and four tribes: Parthenini, Adoliadini (=Euthaliini), Limenitidini, and Neptini (*Wahlberg, 2007*). Limenitidinae are distributed worldwide and occur in all major biogeographical regions: Nearctic, Neotropics, Palaearctic, Afrotropics, Oriental, and Australasia (*Chermock, 1950*; *Chou, 1998*; *Willmott, 2003*). The species of the tribe Parthenini are limited to the Oriental and Australasian regions while the species of the tribes Neptini and Adoliadini are distributed throughout the Old World tropics. The species of the tribe Limenitidini are distributed mainly in the Palaearctic and the New World. It should be noted that some studies (e.g., *Mullen et al., 2011*) have included *Lelecella* as a limenitidine, although this genus is in fact in the subfamily Apaturinae.

Initial studies on Limenitidinae were mostly limited to the description of new species and genera. *Schatz (1892)* studied and classified the Limenitidinae of the world in three tribes ("Neptis-Gruppe", "Limenitis-Gruppe", and "Euthalia-Gruppe") based on venation and palpal structures. Later, *Reuter (1896)* classified Limenitidinae into two tribes: Limenitidi and Neptidi based on studies of the palpi. The tribe Limenitidi (including the Euthalid complex) was further subdivided in two subtribes Limenitini and Parthenini. *Moore (1890)* surveyed the limenitidines of south eastern Asia introducing many new generic names and grouped them into two tribes Euthaliina and Limenitina (*Neptis* included) based on venation and maculation. Moore's Euthaliina is a synonym of Adoliadina described earlier by Doubleday based on the genus *Adolias* (itself a synonym of *Euthalia*). Moore's name Euthaliina has been in common use, as the following narrative shows. *Aurivillius (1898)* also surveyed and grouped the African Limenitidinae under two tribes Neptididi and Nymphalidi. According to *Chermock (1950)* most of species of Limenitidinae (except *Neptis*) can be distinguished from all other nymphalids by the first anal vein of the forewing that is preserved as a short spur at the base of the cubitus. *Chermock (1950)* considered Limenitidinae of the world to belong to one tribe Limenitini based on venation, male genitalia, life histories, maculation, palpal characters, and distribution. Based on egg

morphology and following *Eliot (1978)* and *Harvey (1991)* divided the tribe Limenitidini into four subtribes: Limenitiditi, Neptiti, Partheniti, and Euthaliiti. However, Chou has divided Asian Limenitidinae into five tribes Euthaliini, Parthenini, Neptini, Limenitini, and Chalingini based on morphological characters (*Chou, 1998*; *Zhang et al., 2011*). *Willmott (2003)* suspected that Chalingini does not belong in Limenitidinae based on their unique morphology. In addition to ambiguous higher classification in Limenitidinae, many genera are vaguely defined or supported by few characters (*Willmott, 2003*).

The systematic relationships within Limenitidinae among its major lineages are still unclear. There have been some genus level phylogenetic studies (*Willmott, 2003*; *Mullen, 2006*; *Mullen et al., 2011*; *Van Velzen et al., 2013*; *Ebel et al., 2015*) and some phylogenetic studies included a few genera of the subfamily Limenitidinae (*Zhang et al., 2008*; *Zhang et al., 2011*; *Wu et al., 2014*). However, a comprehensive phylogenetic study of the entire subfamily at the genus and tribe level is still lacking, thus hindering evolutionary studies of the subfamily. Furthermore, a solid phylogenetic hypothesis of Limenitidinae is required to study the evolutionary processes that drive rates of diversification in the subfamily.

Our aims are to study systematics of the subfamily Limenitidinae using up to 18 gene regions per species of 205 taxa belonging to recognized genera and tribes of Limenitidinae spanning all major biogeographical areas. We also introduce seven new gene regions (CycY, Exp1, Nex9, PolII, ProSup, PSb and UDPG6DH) used in this study which have never been previously used for phylogenetic studies.

## MATERIAL AND METHODS

### Taxon sampling

A total of 205 samples representing 39 genera and all four traditional tribes (Table S1): Parthenini, Neptini, Adoliadini and Limenitidini of the subfamily Limenitidinae were collected either by the authors during field visits or by various collaborators. Samples were acquired from all major biogeographical areas. Unfortunately, we could not obtain sequence data from three potentially important genera (*Neurosigma*, *Euryphaedra*, and *Kumothales*). Four exemplar taxa from the sister subfamily Heliconiinae: *Argynnis*, *Heliconius*, *Actinote*, and *Cethosia* were selected as outgroups to root the topology of the subfamily Limenitidinae.

Genomic DNA was mainly extracted from one or two legs, and in a few cases thoracic tissue, of dried mounted vouchers or ethanol-preserved specimens of butterflies. Genomic DNA was extracted using the Qiagen DNEasy extraction kit, following the protocol from the manufacturer. For each species, we amplified and sequenced one gene from mitochondrial genome (*cytochrome oxidase subunit I,* COI) and 17 genes from nuclear genomes, of which *carbamoylphosphate synthetase* (CAD), *Ribosomal Protein S5* (RpS5), *Ribosomal Protein S2* (Rps2), *wingless* (wgl), *cytosolic malate dehydrogenase* (MDH), *glyceraldehydes-3-phosphate dehydrogenase* (GAPDH), *Elongation factor 1 alpha* (EF-1a), *Arginine Kinase* (ArgKin), *Isocitrate dehydrogenase* (IDH) and *dopa-decarboxylase* (DDC) were amplified using primers and protocols from *Wahlberg & Wheat (2008)*. For the new gene regions *Cyclin Y* (CycY), *exportin-1-like* (Exp1), *sorting nexin-9-like* (Nex9), *DNA-directed RNA polymerase II polypeptide* (PolII), *suppressor of profiling 2* (ProSup), *proteasome beta subunit* (PSb),

*UDP glucose6 dehydrogenase* (UDPG6DH) as well as a different section of ArgKin we used primer pairs and protocols described by *Wahlberg et al. (2016)*. For a number of species, sequences were downloaded from GenBank (accession numbers in Table S1).

Successful amplicons were cleaned with A'SAP (ArticZymes) and Sanger sequenced (Macrogen Services, Amsterdam). Previously published DNA sequences (*Wahlberg et al., 2009*; *Mullen et al., 2011*; *Van Velzen et al., 2013*; *Wu et al., 2014*) were also included in the current study. Nucleotide sequence alignment was manually done using the program Bioedit (*Hall, 1999*). Sequences were managed and datasets were constructed using VoSeq v1.7.4 (*Peña & Malm, 2012*).

## Phylogenetic inference

Phylogenetic analyses were done first separately for each gene (producing gene trees) and then for all the 18 genes combined. The combined dataset is given in Data S1. We explored various partitioning schemes of our concatenated multi-gene dataset using PartitionerFinder v1.1.1 (*Lanfear et al., 2012*) and compared them based on the Bayesian Information Criterion (BIC). We first partitioned by gene and codon positions and ran PartitionFinder in order to find which subsets could be combined. In addition, we calculated the relative rates of evolution for each site in the alignment using TIGER (*Cummins & McInerney, 2011*) and created partitions using the RatePartitions algorithm (*Rota, Malm & Wahlberg, 2017*). We tested a range of *d* values (2.0–5.0, with increments of 0.5), which affects the number of partitions, and calculated their BIC values in PartitionFinder.

Phylogenetic inference analyses were carried out using both Maximum likelihood (ML) and Bayesian Inference (BI) methods. Maximum likelihood phylogenetic inference analyses were carried out in RAxML v8.2.4 (*Stamatakis, 2014*) on XSEDE on the CIPRES Science Gateway v3.3 (*Miller, Pfeiffer & Schwartz, 2010*) using the best partition scheme suggested by the PartitionFinder/TIGER analysis based on BIC. For bootstrapping, we performed 1,000 Maximum Likelihood (ML) pseudo-replicates analyses and bootstrapping was performed under auto Majority Rule Criterion (autoMRE). Similarly, BI was performed using Markov Chain Monte Carlo (MCMC) in MrBayes v3.2.6 (*Ronquist et al., 2012*) on XSEDE on the CIPRES Science Gateway v3.3 (*Miller, Pfeiffer & Schwartz, 2010*). Two parallel runs of four chains (three heated and one cold) were performed for 20 million generations, with sampling done at every 1,000th generation. The software Tracer v1.6 (*Rambaut et al., 2014*) was used to inspect the sample sizes of the parameters used in the BI and also check for the convergence or otherwise of the parallel MCMC runs.

As there was a lot of missing data for many specimens (Table S1), we also analysed a subset of taxa that had 10 or more gene regions sequenced. This set of 55 taxa (including all the outgroups) was analysed with RAxML as described above, partitioned by gene.

## Taxonomic decisions

The electronic version of this article in Portable Document Format (PDF) will represent a published work according to the International Commission on Zoological Nomenclature (ICZN), and hence the new names contained in the electronic version are effectively published under that Code from the electronic edition alone. This published work
**Table 1** Basic statistics for each gene region used in this study.

| Data set | Data type | Length (bp) | Dataset completion (%) | Variable (%) | Pars. Inf. (%) | Invariable (%) | Freq. A (%) | Freq. T/U (%) | Freq. C (%) | Freq. G (%) |
|---|---|---|---|---|---|---|---|---|---|---|
| ArgKin | Nuclear | 742 | 25.3 | 33.69 | 28.3 | 66.31 | 24.42 | 19.91 | 30.51 | 25.16 |
| CAD | Nuclear | 850 | 12.2 | 43.29 | 34.94 | 56.71 | 35.23 | 30.71 | 13.83 | 20.23 |
| COI | Mitochondrial | 1,475 | 84.6 | 49.97 | 40.54 | 50.03 | 29.15 | 40.03 | 16.16 | 14.66 |
| CycY | Nuclear | 375 | 24.8 | 36 | 31.47 | 64 | 31.83 | 31.78 | 15.69 | 20.7 |
| DDC | Nuclear | 373 | 14.4 | 44.24 | 38.34 | 55.76 | 25.4 | 28.96 | 24.38 | 21.25 |
| EF1a | Nuclear | 1,240 | 73.8 | 35.97 | 30.65 | 64.03 | 26.75 | 22.42 | 26.5 | 24.34 |
| Exp1 | Nuclear | 729 | 8.6 | 35.25 | 27.57 | 64.75 | 31.76 | 30.58 | 16.47 | 21.18 |
| GAPDH | Nuclear | 691 | 69.1 | 40.23 | 35.31 | 59.77 | 25.16 | 27.26 | 25.25 | 22.32 |
| IDH | Nuclear | 710 | 34.3 | 44.08 | 39.72 | 55.92 | 32.43 | 27.31 | 18.93 | 21.33 |
| MDH | Nuclear | 733 | 18.1 | 32.88 | 20.33 | 67.12 | 28.36 | 27.03 | 21.6 | 23.01 |
| Nex9 | Nuclear | 420 | 25 | 43.57 | 36.43 | 56.43 | 34.33 | 25.8 | 19.42 | 20.45 |
| PolII | Nuclear | 360 | 24.2 | 39.17 | 35.56 | 60.83 | 31.49 | 29.65 | 16.13 | 22.73 |
| ProSup | Nuclear | 432 | 14.1 | 39.35 | 29.4 | 60.65 | 26.84 | 30.72 | 18.46 | 23.99 |
| PSb | Nuclear | 366 | 24.1 | 42.62 | 40.16 | 57.38 | 28.86 | 26.14 | 22.31 | 22.68 |
| RpS2 | Nuclear | 411 | 23.9 | 39.42 | 34.06 | 60.58 | 24.98 | 24.59 | 21.68 | 28.75 |
| RpS5 | Nuclear | 617 | 63.5 | 41.82 | 38.74 | 58.18 | 27.44 | 25.19 | 23.01 | 24.35 |
| UDPG6DH | Nuclear | 405 | 16.9 | 37.78 | 35.8 | 62.22 | 30.15 | 28.79 | 19.9 | 21.17 |
| Wingless | Nuclear | 400 | 78.1 | 53.5 | 42.75 | 46.5 | 23.66 | 19.56 | 27.75 | 29.04 |

and the nomenclatural acts it contains have been registered in ZooBank, the online registration system for the ICZN. The ZooBank LSIDs (Life Science Identifiers) can be resolved and the associated information viewed through any standard web browser by appending the LSID to the prefix http://zoobank.org/. The LSID for this publication is: urn:lsid:zoobank.org:pub:A422503C-2E62-4001-8397-B8C9085CB23C. The online version of this work is archived and available from the following digital repositories: PeerJ, PubMed Central and CLOCKSS.

## RESULTS

### Molecular data

Our final molecular data matrix consisted of 209 taxa representing 205 Limenitidinae species; four related taxa as outgroups; and 11,327 aligned nucleotide sites with no indels. In this study, we used 18 genes of which seven genes (CycY, Exp1, Nex9, PolII, Prosup, PSb and UDPG6DH) have not been previously used in phylogenetic studies of Nymphalidae butterflies. Table 1 gives the basic statistics for variation in each gene region. The new gene regions show similar amounts of variation to the standard gene regions of *Wahlberg & Wheat (2008)*.

The best partitioning scheme was evaluated based on BIC values as calculated by PartitionFinder (*Lanfear et al., 2012*). Partitioning strategies based on genes were decisively worse than those based on RatePartitions or partitioning by gene and codon position (Table 2). The best partitioning scheme was created by RatePartitions with $d = 5.0$, which

**Table 2  BIC scores for the different partitioning strategies as calculated by PartitionFinder.** "PF" means PartitionFinder was allowed to find the optimal strategy with predefined partitions (by gene or by codon position by gene). "TIG" refers to TIGER partitioning using RatePartitions with *d* set to the number given (see *Rota, Malm & Wahlberg, 2017* for details).

| Partitions | BIC | Difference to best |
|---|---|---|
| Partition_18_genes | 334394.0202 | 12525.6029 |
| Partition_PF_gene | 334060.5349 | 12192.11755 |
| LimenTIG2.0_parts | 322668.1567 | 799.739362 |
| LimenTIG4.5_parts | 322615.2355 | 746.818151 |
| LimenTIG4.0_parts | 322546.4603 | 678.042952 |
| LimenTIG3.5_parts | 322518.0154 | 649.598091 |
| LimenTIG2.5_parts | 322517.0803 | 648.662997 |
| LimenTIG3.0_parts | 322498.6547 | 630.237363 |
| Partition_PF_codon | 322411.781 | 543.363707 |
| LimenTIG5.0_parts | 321868.4173 | |

subdivided the data into 19 partitions. This partitioning scheme had a BIC value 543 units lower than the next best scheme based on partitioning by gene and codon position. We thus used the RatePartitions 5.0 scheme for further analyses.

## Systematics

With four outgroups, the maximum Likelihood (ML) (Fig. 1) and Bayesian Inference (BI) (Fig. S1) methods recovered the subfamily Limenitidinae as monophyletic with strong bootstrap supports (BS 100) and high posterior probabilities (PP 1.0). Our analyses recovered seven major lineages: a clade including species of the tribe Parthenini, a clade including *Bhagadatta*, *Harma* and *Cymothoe*, a clade including *Pseudacraea*, a clade including species of the tribe Neptini, a clade including species of tribe Adoliadini, a clade including *Chalinga pratti* and species of the core tribe Limenitidini (*Harvey, 1991*) and finally an independent lineage leading to *Pseudoneptis bugandensis* of the tribe Limenitidini. Most of these clades are strongly supported, the exceptions are the position of *Chalinga* as sister to the core Limenitidini and the sister position of *Bhagadatta* to *Cymothoe* and *Harma*. The relationships of six of the seven lineages are not resolved despite increased gene region sampling, only the sister relationship of Parthenini to the rest of Limenitidinae is strongly supported. Reducing the taxon sampling to only those taxa with 10 or more gene regions sequenced did not change the fundamental results in any way (Fig. S2).

The relationships within the *Cymothoe* clade are very similar to those reported in a previous study of the genus (*Van Velzen et al., 2013*), with the exception of the genus *Bhagadatta* which appears to be sister to *Cymothoe* and *Harma* with low to moderate support (BS 57, PP 0.98).

The genus *Pseudacraea* formed an independent lineage that appears to be sister to Neptini with no support in ML (BS 39) and moderate support in BI (PP 0.98). Relationships of species within *Pseudacraea* were generally well supported and clear, with *P. poggei* and *P. lucretis* being the sister group of the rest of the genus.

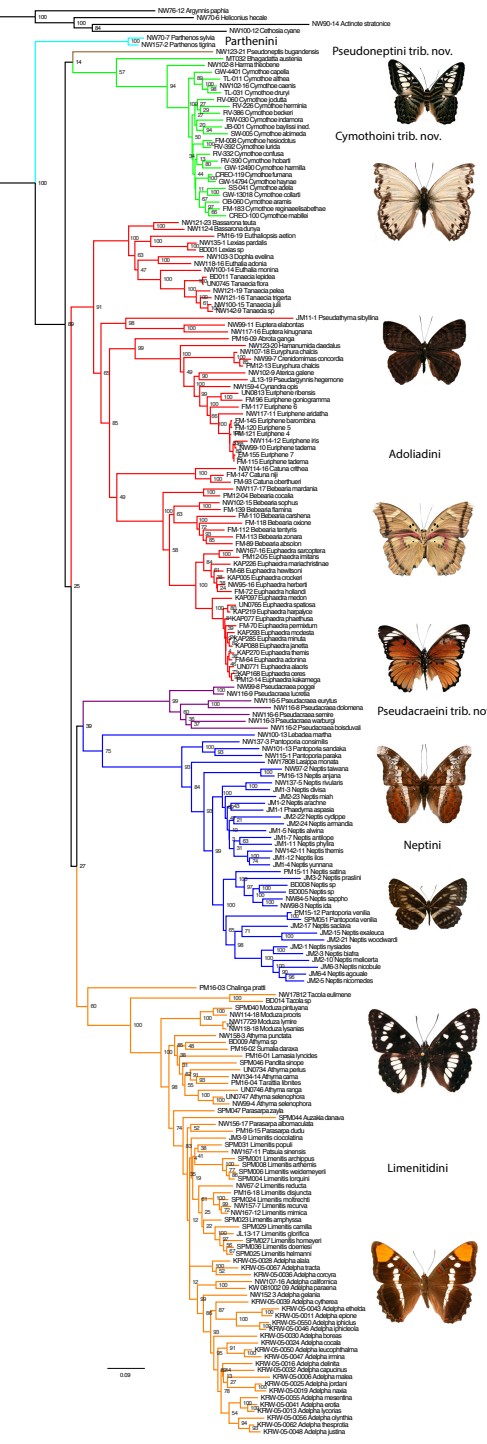

**Figure 1** **The Maximum Likelihood topology for Limenitidinae with associated bootstrap values.** Major lineages that are considered tribes in this paper are coloured. Examples of butterflies (voucher specimens for this work) from top: *Parthenos sylvia*, *Cymothoe caenis*, *Euriphene tadema*, *Euphaedra herberti*, *Pseudacraea poggei*, *Lebadea martha*, *Neptis ida*, *Limenitis reducta* and *Adelpha californica*.

In Neptini, we found the genus *Lebadea* to be sister to the rest of the tribe, with the core *Pantoporia* branching off next and *Lasippa* being sister to *Neptis s.l.* We recovered the genus *Pantoporia* as polyphyletic and *Neptis* as paraphyletic. The species *Pantoporia venilia* from Australia was found to be sister to a clade of African species of *Neptis* with low support values (BS 65, PP 0.53), but certainly within a strongly supported (BS 100, PP 1.0) derived clade of *Neptis*. The species *Phaedyma aspasia* was also found within one of the well supported Asian *Neptis* clades. Asian species of *Neptis* formed a grade while all sampled African species were found in a strongly supported clade (BS 98, PP 1.0).

In Adoliadini, we found five well supported clades, the Asian *Euthalia* clade, and the African *Euptera*, *Hamanumida*, *Catuna* and *Bebearia* clades. Relationships of these five lineages were not well supported, with the African clades forming a monophyletic group in the ML analyses (BS 65), but not in the BI analyses, where the *Euptera* clade was the sister to the rest of Adoliadini (PP 1.0). The Palaearctic species *Abrota ganga* was sister to the *Hamanumida* clade with very high support values (BS 99, PP 1.0) rather than clustering with other Asian Adoliadini. We recovered the genus *Euthalia* as polyphyletic, with *Euthalia adonia* being the sister to *Dophla evelina* with strong support while *Euthalia monina* was sister to species of the genera *Tanaecia* with strong support values. *Bebearia* was found to be paraphyletic with regard to *Euphaedra* with low bootstrap (BS 58) but high posterior probability (PP 0.99). The species *Crenidomimas concordia* was found to be nested within the species *Euryphura chalcis* with all three specimens being genetically very similar.

We found *Chalinga pratti* to be sister to the core Limenitidini with low or no support (BS 60, PP 0.56), but this position was consistent and stable in all analyses. Within the core Limenitidini there are five well supported lineages, with *Tacola* sister to the rest, *Moduza* branching off next, then the *Athyma* clade, and finally *Parasarpa zayla* as sister to the *Limenitis* clade. The genus *Tarattia* was found to be within *Moduza*. The *Athyma* clade comprises the paraphyletic *Athyma* with the genera *Sumalia*, *Pandita* and *Lamasia* deeply within the genus. Also the *Limenitis* clade shows nonmonophyletic genera: *Parasarpa*, *Limenitis* and *Adelpha* are intermixed and the clade contains the genera *Auzakia* and *Patsuia*. Relationships differ somewhat between the ML and BI analyses in this part of the tree, especially where branch lengths are very short or nonexistent.

## DISCUSSION

### Systematic implications

Here, we studied molecular systematics of the recently defined (*Wahlberg et al., 2009*) subfamily Limenitidinae. Previous studies (*Brower, 2000*; *Wahlberg, Weingartner & Nylin, 2003*; *Freitas & Brown, 2004*) clearly showed that the traditional view of the subfamily Limenitidinae (e.g., *Harvey, 1991*) was not monophyletic. *Wahlberg et al. (2009)* defined the subfamily but did not discuss the internal relationships. We recovered seven independent lineages corresponding to four tribes Parthenini, Neptini, Adoliadini, Limenitidini; as well as three independent lineages without formal tribal names: the *Cymothoe* clade, *Pseudoneptis* and *Pseudacraea* (Fig. 1). For consistency, when discussing previous publications, we will align taxon concepts with ours, e.g., our concept of the subfamily Limenitidinae has often been referred to as the tribe Limenitidini, and our tribes as subtribes.

Many of the relationships we found were surprising, but some were anticipated by *Willmott (2003)* based on careful morphological comparisons. For instance he noted similarities in male genitalia between *Lebadea* and *Neptis*, suggested that *Bhagadatta* might be related to *Cymothoe*, that *Tacola* is sister to the rest of Limenitidini, maintained that *Parthenos* is the only genus to be included in Parthenini, and proposed that *Cymothoe* be placed in a tribe of its own. *Willmott (2003)* also suggested that Neptini was not a separate entity from Limenitidini, as did *Amiet (2000)*, whereas based on our analyses it is clearly a separate entity that is not even sister to Limenitidini.

With the exception of the position of Parthenini as sister to the rest of Limenitidinae, the relationships of the major lineages within the subfamily were poorly supported despite up to 18 gene regions being sequenced for specimens within each lineage. The branches subtending these lineages are characterised by very short lengths, suggesting a period of rapid divergences. Such patterns are repeated throughout the evolutionary history of Limenitidinae, notably within *Cymothoe* (*Van Velzen et al., 2013*), *Euriphene*, *Euphaedra* and the base of the *Limenitis* clade.

## Parthenini

As anticipated by *Willmott (2003)*, our data recovered only species of *Parthenos* in this tribe and its position as sister to the rest of Limenitidinae was recovered with strong support in all phylogenetic analyses. Similar results were also found by *Zhang et al. (2008)*, *Zhang et al. (2011)* and *Wu et al. (2014)*. *Parthenos* is limited to the Indo-Australian region.

## Cymothoini Dhungel & Wahlberg *trib. nov.*

LSID urn:lsid:zoobank.org:act:C26A6D77-EDE1-43DB-919F-254E47B82CA3

Based on our results, the genera *Cymothoe*, *Harma* and likely *Bhagadatta* form an independent lineage that warrant tribal status. *Harvey (1991)* classified the two African genera *Harma* and *Cymothoe* in the tribe Limenitidini. However, *Amiet (2001)* and *Willmott (2003)* regarded *Cymothoe* (including *Harma*) as *incertae sedis*, as they share more morphological features with Adoliadini than with Limenitidini. The genera *Harma* and *Cymothoe* were recovered as sister to each other with strong support values. The *Harma* + *Cymothoe* sister clade relationship was consistent with the previous study by *Van Velzen et al. (2013)*. *Harma* and *Cymothoe* are here placed in a new tribe Cymothoini Dhungel & Wahlberg **trib. nov.** The tribe forms a strongly supported clade comprising species placed in *Cymothoe* and *Harma* with DNA sequence data from the following gene regions (exemplar sequences from *Cymothoe caenis*) ArgKin (GQ864537), CAD (GQ864636), COI (GQ864754), CycY (MG741765), DDC (MG741734), EF1a (GQ864848), GAPDH (GQ864952), IDH (GQ865083), MDH (GQ865196), Nex9 (MG741407), PolII (MG741353), ProSup (MG741316), PSb (MG741271), RpS2 (GQ865312), RpS5 (GQ865420), UDPG6DH (MG741133) and wingless (GQ864442).

Surprisingly, we recovered species *Bhagadatta austenia* as a sister to genera *Harma* + *Cymothoe* but with a weak support values (BS 57, PP 0.98). *Bhagadatta austenia* has been classified in the tribe Limenitidini by *Harvey (1991)* and *Wu et al. (2014)* but *incertae sedis* by *Willmott (2003)*, who noted similarities in genitalia with *Cymothoe*. We retain

*Bhagadatta* as *incertae sedis* in Limenitidinae, but suggest that it might be placed in the new tribe Cymothoini once further information is available. Interestingly, *Bhagadatta* is restricted to Asia whereas *Harma* and *Cymothoe* are African genera. Only COI sequences were available for *Bhagadatta* from the study of *Wu et al. (2014)*, thus it is imperative that nuclear genes are sequenced from this taxon to test its position.

### Neptini

Neptini including *Lebadea* was recovered as monophyletic with moderate support (BS 75, PP 0.98). The monotypic genus *Lebadea* was classified as a member of tribe Parthenini by Harvey, but *Willmott (2003)* removed it to Limenitidini and suggested similarities to *Neptis* in male genitalia. *Wahlberg et al. (2009)* found the genus to be sister to Neptini, with no comment, a position that we corroborate here with more data. The core Neptini, including the genera *Neptis*, *Pantoporia*, *Lasippa* and *Phaedyma*, form a strongly supported clade, with *Pantoporia* being sister to *Lasippa* and *Neptis*, and *Phaedyma aspasia* being within *Neptis*. *Phaedyma aspasia* was originally described in *Neptis* by Leech but has been placed in *Phaedyma* by various authors, e.g., *Chou (1998)*. Unfortunately we were not able to sample the type species of the genus *Phaedyma* (*P. heliodora*, synonym of *P. amphion*), thus we are unable to say whether the genus should be synonymized with *Neptis*. We propose a revised combination, *Neptis aspasia* **comb. rev.** Similarly, *Pantoporia venilia* does not belong in the genus *Pantoporia*, but is clearly within *Neptis*, leading to another revised combination *Neptis venilia* **comb. rev**.

The species of *Neptis* are distributed throughout Asia, Africa, Australia, and Europe, with the center of diversity being SE Asia. Our results suggest that the African species form a monophyletic group, with four Asian clades forming a paraphyletic grade with regard to the African clade.

### Pseudacraeini Dhungel & Wahlberg trib. nov.

LSID urn:lsid:zoobank.org:act:E9569B8F-4D9D-4BCC-A18F-431557043079

Our results recovered the genus *Pseudacraea* as a monophyletic group with strong support values, and suggest that *Pseudacraea* might be sister to Neptini, although with no support in ML. *Pseudacraea* has been classified as Limenitidini (*Harvey, 1991*; *Willmott, 2003*). *Amiet (2000)* and *Willmott (2003)* suggest that *Pseudacraea* share synapomorphies with Limenitidini and Neptini, and indeed our ML topology suggests that these three lineages form a monophyletic group, however with no support at all. It appears that *Pseudacraea* is an independent lineage much like *Pseudoneptis* and the *Cymothoe* clade, and is thus placed in a tribe of its own: Pseudacraeini Dhungel & Wahlberg **trib. nov.** The tribe comprises species in the genus *Pseudacraea* and can be characterized by the DNA sequence data from the following gene regions (example from *Pseudacraea poggei*) ArgKin (MG741852), CAD (GQ864704), COI (GQ864802), CycY (MG741798), EF1a (GQ864896), Exp1 (MG741609), GAPDH (GQ865024), IDH (GQ865143), MDH (GQ865258), Nex9 (MG741441), PolII (MG741387), ProSup (MG741336), PSb (MG741302), RpS2 (GQ865362), RpS5 (GQ865489), UDPG6DH (MG741157) and wingless (GQ864490).

## Adoliadini

The monophyly of Adoliadini is strongly supported (BS 91, PP 1.0). This tribe contains species from genera that are distributed in both Asia and Africa. Based on biogeography, Adoliadini could be divided into two subtribes: Adoliadina (*Euthalia* clade) for the Asian and Bebearina (*Hamanumida*, *Bebearia* and *Catuna* clades) for African species. This division does not take into account the African *Euptera* clade, containing the genera *Euptera* and *Pseudathyma*, which does not have a stable position in our analyses, being either sister to all Adoliadini (BI, PP 1.0) or sister to the other African clades (ML, BS 65). This suggests that using the concept of subtribe is not particularly useful in this case. Surprisingly, the Asian genus *Abrota* was sister to the African *Hamanumida* clade with strong support values (BS 99, PP 1.0) rather than clustering with other Asian Adoliadini.

The genus *Euthalia* was recovered as paraphyletic with *Euthalia monina* being sister to *Tanaecia* and *Euthalia adonia* being sister to *Dophla* with strong support values. This pattern is intriguing and calls for a much more detailed study of the species rich genus *Euthalia*. Another intriguing pattern is the genetic similarity of *Crenidomimas concordia* with *Euryphura chalcis*. These two taxa are very different based on wing patterns, with *Crenidomimas* perhaps mimicking the genus *Sevenia* (Nymphalidae: Biblidinae), but clearly they are very closely related to each other and should be the focus of a more detailed study. The genus *Bebearia* was also found to be paraphyletic with regard to *Euphaedra*, although with only moderate support in ML analyses. This clade also requires further study in order to establish whether a new genus needs to be described.

## Limenitidini

The position of *Chalinga pratti* (also known as *Seokia pratti*) as sister to the core Limenitidini was stable across all analyses, but never had high support. As noted in the Introduction, *Chou (1998)* placed *Chalinga* in its own tribe Chalingini and *Willmott (2003)* suspected that *Chalinga* (including *Seokia*) perhaps did not belong to Limenitidinae. Our results show that it does indeed belong to the subfamily, and is likely to be the sister group to the core Limenitidini. For the time being we prefer to keep *Chalinga* in the tribe Limenitidini until there is further evidence that it should be considered a separate lineage worthy of tribal status.

The core Limenitidini comprises five distinct lineages, of which three show para- and polyphyly of constituent genera. These are the *Moduza*, *Athyma* and *Limenitis* clades. In addition, the genus *Tacola* and the species *Parasarpa zayla* form independent lineages. Two species endemic to Sulawesi have been removed from *Moduza* and placed in the genus *Tarattia* (*Hanafusa, 1989*; *Tsukada, 1991*), of which we sampled *T. lysania*. We found *T. lysania* to be sister to *Moduza lymire*, also endemic to Sulawesi, but retained in the genus *Moduza* (*Vane-Wright & De Jong, 2003*). We suggest that until further evidence shows that the Sulawesian clade is clearly sister to *Moduza* and not within it, *Tarattia* should be considered a synonym of *Moduza*. The genus *Athyma* has three relatively small genera within it: *Pandita*, *Sumalia*, and *Lamasia*. *Lamasia lyncides* was separated from *Moduza* by *Tsukada (1991)*, but appears to actually be a species of *Athyma*. As the three genera are well within *Athyma*, they should be synonymized with it.

The phylogenetic relationships of genera within the *Limenitis* clade are complex and unresolved. The type species of the genus *Parasarpa* (*P. zayla*) is an independent lineage sister to the *Limenitis* clade with good support, but other members of the genus are found within the clade in an unresolved position. *Adelpha* is found in two well supported clades that may or may not be sister to each other, a result also found by *Mullen et al. (2011)*. The monotypic *Patsuia* appears to be sister to the type species of *Limenitis* (*L. populi*) and thus the former can be synonymized with the latter genus. The position of the monotypic *Auzakia* varies depending on the method of analysis, with ML placing it as sister to the rest of the *Limenitis* clade, while Bayesian inference places it within *Limenitis*. On the whole, the genus *Limenitis* presents a challenge for classification and clearly more data are necessary to resolve the relationships.

### Pseudoneptini Dhungel & Wahlberg trib. nov.

LSID urn:lsid:zoobank.org:act:AB322712-F361-4FDD-A6C3-E9DEC6EA9402

The genus *Pseudoneptis* was classified in the tribe Limenitidini by *Harvey (1991)* but *incertae sedis* by *Willmott (2003)*. In this study, *Pseudoneptis* is recovered as sister either to the *Cymothoe* clade or to Limenitidini depending on method of analysis, i.e., it is highly unstable. Given that we have sequenced 14 gene regions from our specimen, the instability is more likely to be due to a rapid divergence scenario than a lack of data. This suggests that *Pseudoneptis* should be placed in a tribe of its own, especially since the single species in the genus has a suite of apomorphies (*Amiet, 2002*). We thus erect a monotypic tribe Pseudoneptini Dhungel & Wahlberg **trib. nov.** for the species *Pseudoneptis bugandensis*. Apomorphies for the tribe are described in *Amiet (2002)* and the lineage is also diagnosed by the the unique combination of DNA sequence data from the following gene regions ArgKin (MG741830), CAD (GQ864705), COI (GQ864803), CycY (MG741777), EF1a (GQ864897), GAPDH (GQ865025), IDH (GQ865144), MDH (GQ865259), Nex9 (MG741419), PolII (MG741365), PSb (MG741283), RpS2 (GQ865363), UDPG6DH (MG741142) and wingless (GQ864491).

## CONCLUSION

This study presents the most comprehensive phylogenetic analysis to date for the "trash-can" subfamily Limenitidinae. Based on fragments of up to 18 genes per species, 205 species and four outgroups, our results recovered Limenitidinae as a monophyletic clade and which comprises seven major lineages that deserve tribal status. Four tribes have been traditionally recognized: Parthenini, Neptini, Adoliadini, and Limenitidini, while three lineages are placed in new tribes here: Cymothoini, Pseudoneptini and Pseudacraeini. The new Cymothoini tribe includes two African genera *Cymothoe* and *Harma*, and quite likely an Asian genus *Baghadatta*. The latter two new tribes are monogeneric. At the genus level, we found several traditionally recognized genera to be either poly- or paraphyletic, i.e., *Neptis, Euryphura, Pantoporia, Athyma, Parasarpa, Limenitis*, and *Adelpha*. Further work increasing the taxon sampling is necessary to test the monophyly of these genera and revise their limits.

## ACKNOWLEDGEMENTS

We are grateful to the late Torben Larsen, Zdenek Fric, Freerk Molleman, Kwaku Aduse-Poku, Steve Collins and the African Butterfly Research Institute for providing specimens for this work. We thank Pavel Matos-Maravi, Jenni Mäkynen and Evelyn Sanchez for help in the lab. We thank Martin Weimers and an anonymous referee for comments on a previous version of the manuscript.

### Funding

This study has been funded by an Erasmus Mundus grant to Bidur Dhungel, and grants from the Academy of Finland (Grant No. 265511) and the Swedish Research Council (Grant No. 2015-04441) to Niklas Wahlberg. The funders had no role in study design, data collection and analysis, decision to publish, or preparation of the manuscript.

### Grant Disclosures

The following grant information was disclosed by the authors:
Erasmus Mundus.
Academy of Finland: 265511.
Swedish Research Council: 2015-04441.

### Competing Interests

The authors declare there are no competing interests.

### Author Contributions

- Bidur Dhungel performed the experiments, analyzed the data, wrote the paper, reviewed drafts of the paper.
- Niklas Wahlberg conceived and designed the experiments, performed the experiments, analyzed the data, contributed reagents/materials/analysis tools, wrote the paper, prepared figures and/or tables, reviewed drafts of the paper.

### DNA Deposition

The following information was supplied regarding the deposition of DNA sequences:
   The new sequences generated for this study are available as a nexus file in the Supplemental Material. All sequences used in this study are available on GenBank (new sequences accession numbers MG741008–MG741957).

### Data Availability

   New sequences are available in NCBI GenBank (accession numbers MG741008–MG741957).

### New Species Registration

The following information was supplied regarding the registration of a newly described species:

Publication LSID: urn:lsid:zoobank.org:pub:A422503C-2E62-4001-8397-B8C9085CB23C;

Cymothoini: urn:lsid:zoobank.org:act:C26A6D77-EDE1-43DB-919F-254E47B82CA3;

Pseudacraeini: urn:lsid:zoobank.org:act:E9569B8F-4D9D-4BCC-A18F-431557043079;

Pseudoneptini: urn:lsid:zoobank.org:act:AB322712-F361-4FDD-A6C3-E9DEC6EA9402.

### Supplemental Information

Supplemental information for this article can be found online at http://dx.doi.org/10.7717/peerj.4311#supplemental-information.

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
