# Peer review of "Molecular systematics of the subfamily Limenitidinae (Lepidoptera: Nymphalidae)"

_PeerJ, doi:10.7717/peerj.4311_

## Round 0.1 · original submission · Minor Revisions

· Academic Editor

Minor Revisions

Dear authors

Our reviewers liked your work but suggest some minor changes.

Thanks for submitting your ms to our Journal

Best regards
Michael Wink
Academic editor

·

Basic reporting

This is an excellent study on the molecular systematics of the subfamily Limenitidinae, which is clearly of top standard, even using a large number of nuclear genes never used before in phylogenetic studies of butterflies.

The article is generally written in excellent English, with only a few minor mistakes, which are listed at the bottom.

Extensive background information on the taxonomy of the subfamily is provided including comprehensive referencing.

The article structure is professional and the tables and figures contain all important information in a readable way.

Raw data are shared, but the GenBank accession numbers for new sequences are still missing and need to be added before publication, both in the text (e.g. lines 274-277, 316-320, 383-385) as well as in the supplementary file.

Minor language issues:

line 21: "was" should read "were"
line 68: "Gruppe" means "group" in German and is not necessarily equivalent with a tribe. I suggest to place the group names in quotation marks: ... in three tribes ("Neptis-Gruppe", "Limenitis-Gruppe", and "Euthalia-Gruppe") based on ...
line 172: delete the word "and"
line 301: delete the word "the" before "Asia"
line 387: Change: "Here, the study" to "This study", and change "till date" into "to date"
line 388: change "Based on up to 18 genes regions" to "Based on regions of up to 18 genes"
line 389: add "clade" after "monophyletic"; change "and comprised" to "which comprises"
lines 394-396: revise sentence, e.g.: "At the genus level, our results showed some of the traditionally
recognized genera as either polyphyletic (Neptis and Euryphura) or paraphyletic (Pantoporia, Athyma,
Parasarpa, Limenitis, Adelpha)."
line 397: delete "of the para/polyphyletic nature of the"

Experimental design

The article contains original research and fills an important knowledge gap. Methods are sound, described with sufficient detail and are of very high standard.

Validity of the findings

Robustness of results is checked and taken into account for their interpretation. Conclusions are only drawn for significant findings supported by the results.

Reviewer 2 ·

Basic reporting

The article appears to be well-written, clearly articulated, and well-presented.

Experimental design

The primary limitation of this study is the patchiness of the available data for each species. However, given the taxonomic breadth of the species sampled, and the heavy and necessary reliance upon previously published molecular data, this is unavoidable. Overall, the authors have done an excellent job tackling an unusually difficult taxonomic quandary.

Validity of the findings

Suggestions for improvement. #1) Include a verbal discussion of missing data in the methods and results section. #2) Include a supplemental tree figure that includes only core taxa for which data is available for all gene regions. Such a tree, if possible, would strengthen the confidence that the topological uncertain (especially among the Limenitidini) arises as a consequence of rapid divergence among these genera vs. a lack of phylogenetic resolution due to missing data.

Comments for the author

This phylogenetic investigation of the subfamily Limenitidinae is long overdue. The authors did an excellent job of describing both the previous history of taxonomic confusion related to this group but also placing their new results in a clear and well argued framework. Although additional work will need to be done in the future to fully resolve the generic relationships among the Limenitidini, the current study represent a vast improvement of our understanding of the phylogenetic history of this diverse and complex butterfly lineage.

---

## Round 0.2 · accepted · Accept

· Academic Editor

Accept

Dear Authors

Congratulations- your revision is accepted.

Greetings

Michael Wink
Academic editor